# Static loading of the knee joint results in modified single leg landing biomechanics

Michael W. Olson[1,2]*

**1** Department of Kinesiology, Southern Illinois University Carbondale, Carbondale, IL, United States of America, **2** Department of Athletic Training and Exercise Physiology, Midwestern State University, Wichita Falls, TX, United States of America

* michael.olson@msutexas.edu

## Abstract

### Background

External loading of the ligamentous tissues induces mechanical creep, which modifies neuromuscular response to perturbations. It is not well understood how ligamentous creep affects athletic performance and contributes to modifications of knee biomechanics during functional tasks.

### Hypothesis/purpose

The purpose of this study was to examine the mechanical and neuromuscular responses to single leg drop landing perturbations before and after passive loading of the knee joint.

### Methods

Descriptive laboratory study. Male (n = 7) and female (n = 14) participants' (21.3 ± 2.1 yrs., 1.69 ± 0.09 m, 69.3 ± 13.0 kg) right hip, knee, and ankle kinematics were assessed during drop landings performed from a 30 cm height onto a force platform before and after a 10 min creep protocol. Electromyography (EMG) signals were recorded from rectus femoris (RF), vastus lateralis (VL), vastus medialis (VM), semimembranosus (SM), and biceps femoris (BF) muscles. The creep protocol involved fixing the knee joint at 35˚ during static loading with perpendicular loads of either 200 N (males) or 150 N (females). Maximum, minimum, range of motion (ROM), and angular velocities were assessed for the hip, knee, and ankle joints, while normalized EMG (NEMG), vertical ground reaction forces (VGRF), and rate of force development (RFD) were assessed at landing using ANOVAs. Alpha was set at 0.05.

### Results

Maximum hip flexion velocity decreased (p < 0.01). Minimum knee flexion velocity increased (p < 0.02). Minimum knee ad/abduction velocity decreased (p < 0.001). Ankle ROM decreased (p < 0.001). aVGRF decreased (p < 0.02). RFD had a non-significant trend (p = 0.076). NAEMG was significant between muscle groups (p < 0.02).

**Data Availability Statement:** All relevant data are within the manuscript and its Supporting Information files

**Funding:** The author received no specific funding for this work.

**Competing interests:** The authors have declared that no competing interests exist.

## Conclusion

Distinct changes in velocity parameters are attributed to the altered mechanical behavior of the knee joint tissues and may contribute to changes in the loading of the leg during landing.

## Introduction

Knee joint injuries greatly affect athletic and recreational sport populations. Sex-related and sports based factors are the leading determinants of knee injuries [1,2,3,4]. The loading of the knee joint in dynamic sporting activities influences the stresses and strains, which the tissues within and surrounding the knee joint capsule tolerate. The ligaments of the knee joint provide structural integrity to the joint during both passive and active movements [5]. The incidence of injuries to the passive viscoelastic tissues, such as the anterior cruciate ligament (ACL), posterior cruciate ligament (PCL), medial collateral ligament (MCL), lateral collateral ligament (LCL), and menisci are well documented in the literature [6,4,7] as well as the predictive factors leading to injury [8,9].

The injury mechanisms at the knee joint are multifactorial and are complicated due to the requirements of specific movement activities performed in dynamic environments (i.e., athletics venues). Both contact and non-contact activities play critical roles in determining how the knee joint responds to the given loading conditions. The type of training, task, fatigue level of the individual, and anatomic structure all contribute to the potential injury of the knee joint [10,11]. In particular, when landing from a given height, the knee biomechanics are modified to absorb energy to reduce the impact of the contact forces upon the lower extremities [12]. Females are reported to have greater dynamic knee valgus–a potential sign of knee injury at landing–compared to males, and thus a greater potential for knee injury [13,14,15]. Training individuals to land without excess knee valgus has been documented and may contribute to reduced knee joint injuries [16,17]. Nilstad et al.[18] used static laxity of the ligamentous tissues as a predictor variable for knee valgus, but could not conclude this was a factor responsible for increasing likelihood of knee injuries. Others, however, have determined the laxity of the knee joint ligaments contribute to modifications of the neuromuscular control of the knee joint [19,20,21]. The musculotendinous units contribute much more to the stability of the knee joint during dynamic tasks as the forces generated via the muscle are actively engaged in movement of the joint, as well as resistance to external forces acting upon the knee [22].

Neuromuscular fatigue of the muscles surrounding the knee joint and other lower extremity muscles is a contributing factor to knee injury. Lower extremity kinematics and myoelectric activity, collected with electromyography (EMG), significantly change at landing from a jump or drop after fatigue has been induced [23]. When the muscles become fatigued the ability to generate force diminishes, and the internal moments have reduced capacity to resist the external moments applied [24]. The contribution of neuromuscular fatigue to knee joint injury is significant [11], however, it is unclear how the laxity of the ligamentous tissues contributes to the inability of the joint to maintain its integrity [18]. Neuromuscular fatigue requires the passive ligamentous tissues to be strained further to compensate for the deficiencies of the neuromuscular control. These ligamentous tissues are further loaded during the dynamic activities as tension-relaxation or mechanical creep are induced in parallel with neuromuscular fatigue [25]. Further assessment of the contribution of the passive ligamentous tissues during loading tasks is necessary to understand the mechanisms of injury.

Reductions in both force generating capacity and myoelectric activity of the musculature about the knee joint have been documented during isometric knee actions following passive

loading of the knee joint capsule [20,21,26]. These tests serve to isolate the knee joint and provide a foundational understanding of the influence of loading schemes to the response of the neuromuscular and musculoskeletal systems. Functionally, little is known how passive loading of the knee joint capsular tissues affects the movement of the joint during dynamic (athletic) activities.

The purpose of this study was to examine the mechanical and neuromuscular responses of the lower extremities to landing perturbations before and after passive loading of the knee joint capsule. It was hypothesized that passive static loading of the knee joint capsule would elicit a reduced myoelectric amplitude response from the surrounding musculature at landing. Further, it was also hypothesized that joint kinematics of the landing leg would compensate for the passive loading applied at the knee joint capsule during landing through changes in kinematics parameters.

## Methods

### Participants

University students enrolled in kinesiology classes volunteered to participate in this study. Male (n = 7) and female (n = 14) participants (21.3 ± 2.1 years old, 1.69 ± 0.09m in height, and had a mass of 69.3 ± 13.0 kg) were required to be healthy individuals with no medical conditions which would prevent physical activity, involved in regular physical activity (recreational activities at least 3 times/week), not have any trunk or lower extremity disorders, not have an injury to the head, trunk, and lower extremities within the previous 12 months of participation, and if female not be pregnant. The study was approved by the Southern Illinois University Carbondale Human Subjects Committee (#15277). Participants were provided an informed consent document and written voluntary consent was provided by each participant. Additional verbal instructions were provided during the study. Participants were informed that they could withdraw without penalty at any time during the study.

### Instrumentation

A 6-camera motion capture system (Qualisys, Gotenborg, Sweden) with Oqus 100 camera sampling at 120 Hz was used to collect movement data. Palpation was used to place individual reflective marker spheres of 14 mm diameter bilaterally over the acromion processes, posterior superior iliac spines (PSIS), and anterior superior iliac spines (ASIS). Unilateral markers were positioned over the sacrum at the S1 Table, and the right leg at the greater trochanter, lateral femoral epicondyle, medial femoral epicondyle, lateral malleolus, medial malleolus, calcaneus, 1$^{st}$ and 5$^{th}$ metatarsophalangeal joints. Two four-marker clusters were position on the right leg at the midline of the lateral thigh and the proximal third of the lower leg and secured with Coband © wrapping tape.

Surface electromyography (EMG) (Motion Lab System, Baton Rouge, LA, USA) was used to collect muscle activity from the right thigh musculature surrounding the knee. The skin was abraded and then cleaned with isopropyl alcohol. The 0.02 m diameter stainless steel electrodes have a fixed center to center distance of 0.02 m and bipolar configuration, and were positioned distal to the motor point of each muscle group and aligned parallel with the muscle fibers. Myoelectric signals were collected from the muscles rectus femoris (0.10 m distal from the right ASIS), vastus lateralis (0.10 m proximal and lateral form the patella), vastus medialis (0.10 m proximal and medial to the patella), semimembranosis (~0.20 m proximal to the medial femoral epicondyle), and the biceps femoris (~0.20 m proximal to the lateral femoral epicondyle) [27,26]. Surface EMG signals were bandpass filtered at 20–500 Hz with a common

mode rejection ratio of > 100 dB at 60 Hz, an input impedance of > 100MΩ, and collected at 1200 Hz.

Kinetic data were collected with a 6 degree of freedom force platform (OR-6, AMTI, Watertown, MA, USA) with dimension 0.45 m x 0.5 m embedded and flush with the laboratory floor. Force data were collected at 1200 Hz. Kinematic, EMG, and kinetic data were collected using the Qualisys Tracking Manager (QTM) software interfaced with a USB 2533 12 bit A/D board (Measurement Computing, Inc., Norton, MA, USA) and save for future processing.

## Protocol

Participants warmed up by walking on a motorized treadmill at their self-selected speed for 10 min. Kinematic markers and EMG electrodes were placed upon the participants after the warm-up. Participants performed single leg drop landings from a height of ~ 0.30 m using the right leg (Fig 1). Leg dominance was determined by asking the participants which leg they would use to kick a ball. All participants indicated right leg dominance. Participants began by standing on two legs on top of a box situated 0.10 m horizontal from the force platform. They were instructed to lean forward leading with the right leg in order to drop onto the force platform. Once they landed, the participants were instructed to maintain their one-legged stance and stand erect for 5 s. The hands were positioned on the iliac crests to control arm movements. Participants were given up to 10 practice trials to acclimate to the drop landing task. Participants performed up to 10 trials of drop landings before and after the knee joint capsule was loaded. At least 1 min of rest was provided between trials to reduce the influence of fatigue. Landing trials where the participants either jumped, stepped down, or could not maintain balance at landing were discarded and additional trials were performed. Of the 10 trials recorded, the middle 5 trials were used for analysis.

After the initial drop landing trials, participants were positioned into a high-back chair of a Biodex system 3 dynamometer (Shirley, NY, USA). Participants were positioned with their trunk in an upright erect position with the hips in 90° flexion. Straps were placed across the chest and proximal thigh to reduce movement during the exercise. Then an attachment arm was secured to the dynamometer axis, which was aligned with the lateral femoral epicondyle of the right leg. The attachment arm was secured to the leg 0.05 m proximal to the lateral malleolus. Ramped maximal voluntary isometric efforts (MVIE) of 5 s were performed 3 times each

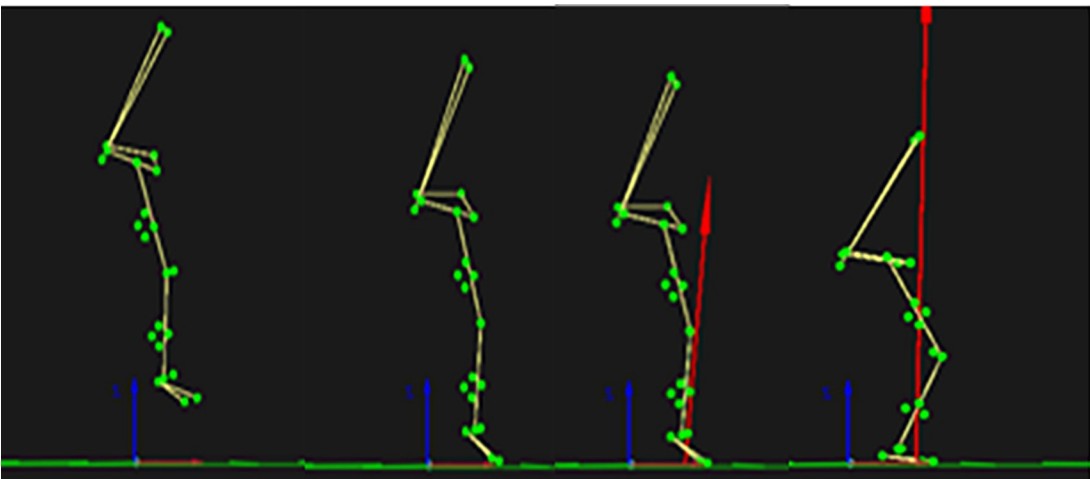

**Fig 1. Drop landing sequence.** An example of the sequence in the drop landing protocol.

with the knee at 90˚ flexion for extension trials and 45˚ flexion for knee flexion trials (full knee extension is 0˚), respectively, with a 60 s rest between efforts. A 10 min rest period was performed after the last MVIE. The leg was then positioned so that the knee was flexed to 35˚ with reference to the anatomical position [20]. A cuff was secured 0.03 m distal to the femoral epicondyle and surrounded the proximal leg. A pulley system was configured to allow a cable to fit perpendicular to the leg and around the cuff. The cable was used to pull the leg anterior relative to the femur with a load of either 200 N (men) or 150 N (women) (Fig 2). This protocol has been reported to increase laxity of the tissues surrounding the knee joint capsule and potentially the ligaments within the capsule [20]. The knee joint was statically loaded for 10 min. Surface EMG was used to ensure a low level of muscle activity relative to the MVIE was maintained during knee loading. Immediately after the loading protocol was completed, participants performed additional drop landing trials.

## Data processing

The EMG signals collected during the static loading were centered, full wave rectified, and low pass filtered at 3 Hz with a fourth order zero-lag Butterworth filter. The EMG signals collected during drop landing trials were centered, rectified, and then low pass filtered at 5 Hz with a fourth order zero-lag Butterworth filter. All EMG data were then normalized to the maximum EMG value attainted during MVIE.

Force data were processed with a low pass Butterworth filter set at 60 Hz using the Motion-Monitor System software (MotionMonitor, Chicago, IL, USA). A vertical ground reaction force (VGRF) threshold of 20 N was establish to determine the onset of load acceptance at landing. Force data were reduced to 120 Hz to coincide with the kinematics data.

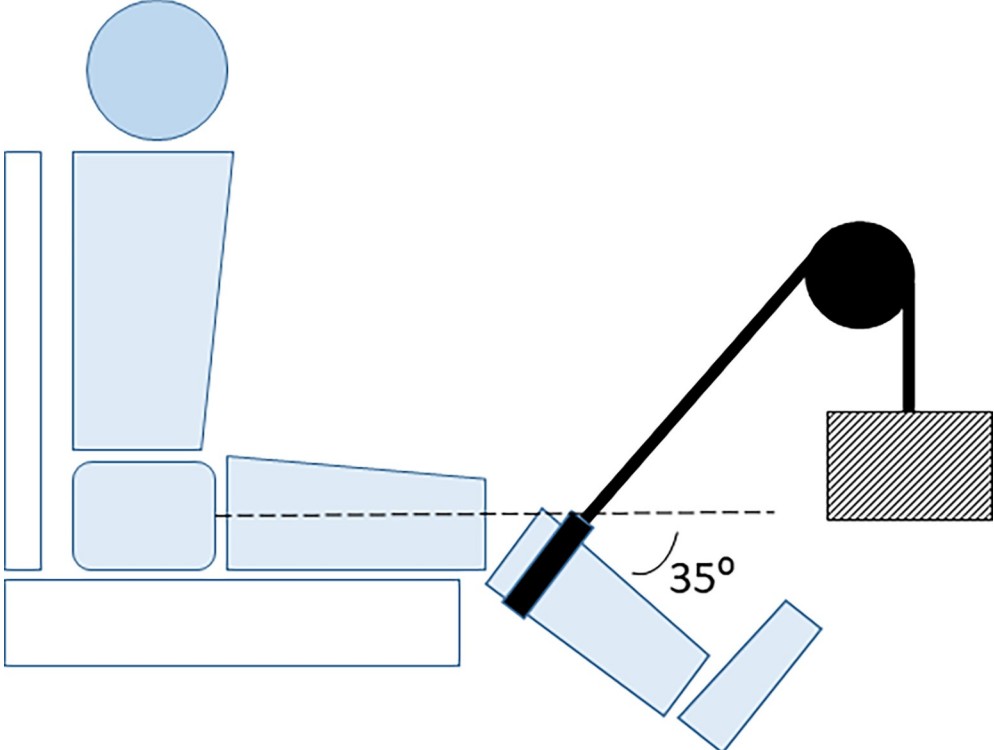

**Fig 2. Set up for the static loading protocol.** Depiction of the static loading of the knee for each participant. The striped box represents the load.

Kinematics data were processed using the MotionMonitor software (MontionMonitor, Chicago, IL, USA). Kinematics data were smoothed with a fourth order low pass zero-lag Butterworth filter set at 10 Hz. A static reference file was used to determine lower extremity segment and joint angles using the right-hand configuration (x-anteroposterior, y-mediolateral, z-vertical axes, respectively). Based upon the static reference file, segmental coordinate systems were established and used in determining relative angles of adjacent segments. Further, Euler angle calculations were performed to determine segment orientations, which contributed to joint angular rotations (X–frontal plane, Y–sagittal plane, and Z–transverse plane) of the distal segments relative to the proximal segments.

## Data analysis

Kinematics data were evaluated in all three planes of movement during the landing phase of the drop landing. Landing phase during the drop landing was assessed as the onset of the force platform threshold until the maximum knee flexion angle was attained. Joint angle parameters (maximum, minimum, and range of motion) were assessed using the right-hand rule as follows: at the knee joint, flexion-extension (about the y-axis), internal-external rotations (about the z-axis), and mediolateral rotations (about the x-axis). At the hip joint, hip flexion-extension (about the y-axis) and hip adduction/abduction (about the x-axis) were evaluated. Ankle plantarflexion and dorsiflexion movements were assessed as these movements have been reported to align with maximum knee joint flexion at landing [28]. Angular velocities (maximum and minimum) of the three knee joint rotations were also calculated to determine differences between single leg landing trials before and after static knee loading.

Kinetic variables of concern were the average and maximum VGRF, force profile of the first 200 ms of landing, and the rate of force development (RFD). Average VGRFs were compartmentalized into four 50ms time intervals (0–50, 51–100, 101–150, and 151–200 ms) to assess the pattern of forces at landing. The RFD was calculated as the difference in maximum VGRF and the VGRF at landing divided by the time between the maximum VGRF and VGRF at landing.

Surface NEMG recordings during the static knee loading protocol were averaged the first 30 s of each minute of the 10 min loading period. This was performed to ensure minimal muscle activity during the loading. Additionally, processed NEMG signals were assessed during the first 200 ms of the landing phase, as well as 200 and 100 ms prior to initiation of the landing to compare landing before (pre-loading landing condition) and after (post-loading landing condition) static knee loading. Pre-and post-loading landing condition NEMG signals were used to determine the preparation of the muscles surrounding the knee joint to the landing (feed-forward control at 200 and 100 ms). The 200 ms after landing was compartmentalized into four 50 ms intervals (0–50, 51–100, 101–150, and 151–200 ms) which were used to average the NEMG signals for evaluating the trend of the neuromuscular activity at landing.

## Statistical analyses

All statistical testing was performed with SPSS v 22.0 (Chicago, IL, USA). Angular displacement and velocity variables (minimum, maximum, and range of motion) were analyzed using a one-way (condition) analysis of variance (ANOVA). A one-way ANOVA was performed to assess average muscle activity at each minute of static loading. Average and maximal NEMG values during landing were analyzed with a 2 factor (condition x muscle) ANOVA. A three-factor, muscle x condition x time interval (5 x 2 x 6) ANOVA was performed on the average NEMG, which included analyzing the pre-landing times at 200 and 100 ms. Average and maximal forces and RFD data were each analyzed with a one-way ANOVA (condition). A 2-way

ANOVA (condition x time interval) was used to assess average VGRF data during the landing phase, while one-way ANOVAs were used to compare maximal VGRF values between conditions. Tukey post-hoc comparisons were performed when significant effects were present. A Mauchly's test of Sphericity was performed to assess the normality of the data. A Greenhouse–Geisser test was applied when normality was not attained. The level of significance was set at $p \leq 0.05$.

## Results

### Electromyography

Average NEMG values did not significantly change during the 10 min of static loading. The average activity for each minute was under 5% of the MVIE, indicating minimum active neuromuscular response to the external load (Table 1).

**Overall average and maximum NEMG during landing.** Average NEMG values did not change between conditions ($p > 0.55$), but were significant between muscles ($F_{4,199} = 5.347$, $p < 0.01$). There was no significant condition x muscle interaction effect ($p > 0.87$) (Table 2). Maximal NEMG values were not significant between conditions ($p > 0.34$), but were significant between muscles ($F_{4,180} = 9.553$, $p < 0.01$) (Table 2). There were no significant interaction effects present ($p > 0.97$).

A significant time interval x muscle interaction was present ($F_{20, 1189} = 1.951$, $p < 0.01$) when observing the average NEMG signals at landing. Post-hoc analysis indicated a significant difference between time intervals ($p < 0.001$) and muscle groups ($p < 0.001$). There were no significant condition effects ($p > 0.27$), nor condition x time interval ($p > 0.99$) or condition x muscle interaction ($p > 0.99$) effects present (Fig 3).

### Kinematics

Maximum and minimum angular displacement data from the hip, knee, and ankle joints are provided in Table 3. Although not significant, a trend was present between conditions for ankle flexion maximum ($p < 0.071$). All other maximum and minimum angular displacement data were not significant between conditions.

**Table 1. Average NEMG from static loading.**

| Time (min) | Muscle Group (% MVIC) | | | | |
|---|---|---|---|---|---|
| | RF | VL | VM | SM | BF |
| 1 | 3.72 (3.8) | 4.15 (2.6) | 3.59 (4.1) | 4.87 (4.3) | 2.80 (2.0) |
| 2 | 3.67 (4.2) | 3.79 (2.0) | 2.67 (1.4) | 4.46 (4.0) | 3.45 (2.7) |
| 3 | 3.57 (4.3) | 3.92 (2.5) | 2.78 (1.5) | 4.06 (3.4) | 3.09 (2.5) |
| 4 | 2.99 (3.2) | 4.08 (2.4) | 2.73 (1.5) | 3.52 (2.5) | 2.82 (2.3) |
| 5 | 3.00 (3.2) | 4.03 (2.7) | 2.73 (1.5) | 3.71 (2.9) | 2.71 (2.4) |
| 6 | 2.95 (3.1) | 3.74 (2.1) | 2.66 (1.4) | 3.49 (2.5) | 2.56 (2.2) |
| 7 | 2.79 (2.9) | 4.11 (2.9) | 2.60 (1.4) | 3.53 (2.5) | 2.79 (2.3) |
| 8 | 2.76 (2.9) | 3.90 (2.1) | 2.89 (1.5) | 3.49 (2.5) | 2.57 (2.1) |
| 9 | 2.75 (2.9) | 4.25 (2.8) | 2.71 (1.5) | 3.43 (2.4) | 2.50 (2.0) |
| 10 | 2.74 (3.0) | 3.84 (1.8) | 3.06 (2.3) | 3.32 (2.4) | 2.34 (1.8) |

Mean (± sd) of normalized surface electromyography as a percentage of MVIC from the three quadriceps and two hamstring muscles during static loading of the knee joint capsule. RF = rectus femoris; VL = vastus lateralis; VM = vastus medialis; SM = semimembranosus; BF = biceps femoris.

**Table 2. NEMG from thigh muscles during pre- and post-loading landing conditions.**

|  | Maximum NEMG | | Average NEMG | |
| --- | --- | --- | --- | --- |
| Muscle | Pre | Post | Pre | Post |
| RF* | 1.15 (0.62) | 1.18 (0.55) | 0.451 (0.30) | 0.477 (0.28) |
| VM^ | 0.43 (0.45) | 0.45 (0.55) | 0.284 (0.32) | 0.241 (0.23) |
| VL† | 0.98 (0.62) | 1.04 (0.50) | 0.506 (0.37) | 0.619 (0.55) |
| BF | 0.87 (0.56) | 0.97 (0.62) | 0.385 (0.28) | 0.391 (0.28) |
| SM‡ | 0.64 (0.31) | 0.809 (0.56) | 0.307 (0.15) | 0.342 (0.29) |

Mean (sd) maximal and average NEMG as a percentage of MVIC from the three quadriceps and two hamstring muscles during pre- and post-loading landing conditions.

*indicates RF significantly greater than VM ($p < 0.01$)

†indicates VL significantly greater than VM, BF, and SM muscle groups (all $p < 0.02$)

^indicates VM significantly less than all other muscle groups (all $p < 0.03$)

‡Indicates SM significantly different than RF, VL, and VM (all $p < 0.03$)

Table 4 provides angular velocity data measured from the hip, knee, and ankle joints. Hip abduction maximum velocity was significant between conditions ($F_{1,3} = 43.5$, $p < 0.007$). A significant difference between conditions was present ($F_{1,9} = 7.963$, $p < 0.02$) for minimum knee flexion velocity. Minimum knee abduction velocity was significant between conditions ($F_{1,9} = 19.35$, $p < 0.002$). All other angular velocities in sagittal, frontal, and transverse planes were not statistically different between conditions.

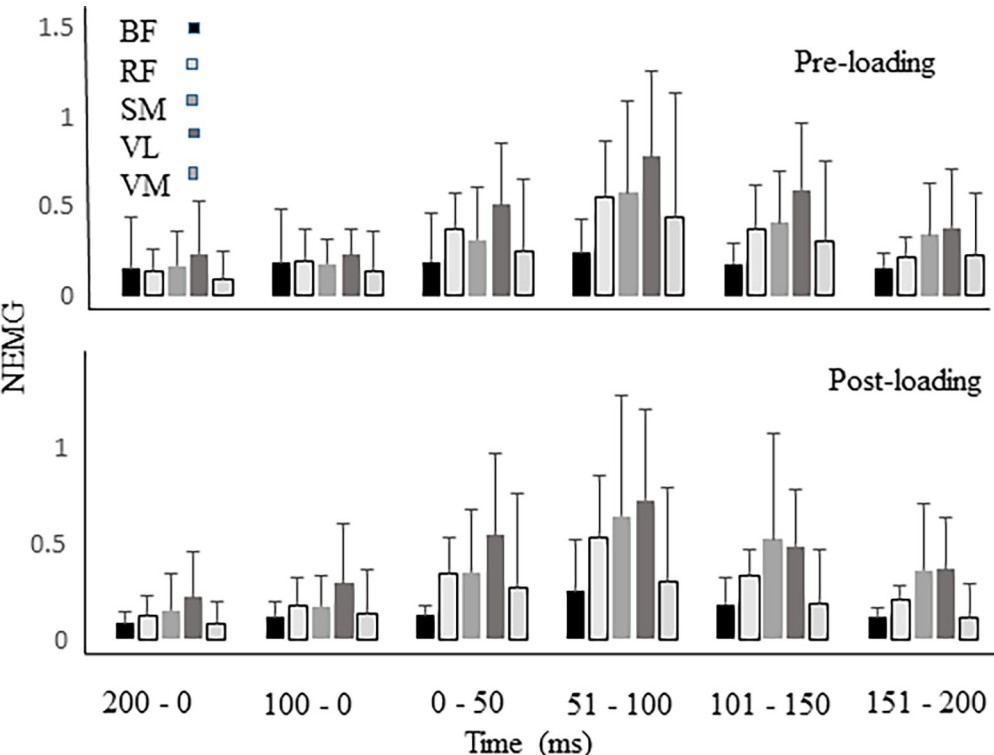

**Fig 3. Average NEMG at 200ms prior to and at landing.** Mean (sd) average NEMG from pre and post-loading landing conditions for rectus femoris (RF), vastus lateralis (VL), vastus medialis (VM), biceps femoris (BF), and semimembranous (SM) muscle groups. NEMG activity is provided in 50ms intervals during the first 200ms of landing. NEMG values are also provided 200ms and 100 ms prior to the landing.

**Table 3. Joint angular displacements at landing.**

| | Maximum | | p-value | Minimum | | p-value |
|---|---|---|---|---|---|---|
| | Pre | Post | | Pre | Post | |
| Hip flexion | 22.0 (6.8) | 19.7 (9.2) | 0.38 | 9.2 (3.0) | 6.2 (6.6) | 0.15 |
| Knee flexion | 60.8 (7.8) | 60.9 (8.3) | 0.97 | 16.4 (4.8) | 15.7 (9.7) | 0.67 |
| Ankle flexion | 1.9 (16.9) | -1.9 (12.6) | 0.071 | -52.7 (14.5) | -51.4 (15.4) | 0.61 |
| Hip Abduction | 14.0 (8.9) | 11.3 (8.6) | 0.13 | -7.6 (13.9) | -6.4 (11.0) | 0.59 |
| Knee abduction | -4.0 (8.3) | -4.2 (6.2) | 0.94 | -30.5 (11.9) | -32.7 (14.1) | 0.29 |
| Knee rotation | 16.2 (5.8) | 16.7 (9.5) | 0.77 | 1.7 (6.9) | 2.8 (2.4) | 0.48 |

Mean (sd) maximum and minimum angular displacements (˚) at the hip, knee, and ankle joints during pre- and post-loading landing conditions. Ankle dorsiflexion is represented as positive, and plantarflexion is noted as negative.

Table 5 provides ROM data for the hip, knee, and ankle joints. Ankle ROM was significant between conditions ($F_{1,166} = 7.904$, $p < 0.006$). No other ROM variables were significantly different between conditions.

## Kinetics

Maximum VGRFs were not different between pre-loading and post-loading landing conditions (1550.5 ± 84.6 N vs.1548.1 ± 55.8 N, $p > 0.91$). The average VGRF measures over the first 200ms of contact at landing were significant between pre-loading and post-loading landing conditions (1297.1 ± 392.4 vs. 1231.3± 392.4 N; $F_{1,795} = 5.593$, $p < 0.018$). A significant difference in the maximum VGRFs between time intervals was present (0-50ms: 940.8 ± 322.7; 51-100ms: 1661.4 ± 491.2; 101-150ms: 1355.5 ± 388.7; 151-200ms: 1099.7 ± 350.2 N) ($F_{3,787} = 128.217$, $p < 0.001$) (Fig 4). Post-hoc comparisons indicated the VGRF was significantly different between each time interval ($p < 0.001$). No significant condition x time interval interaction effect was present ($p > 0.56$). The rate of force development was not significantly different between conditions, but a trend was present (pre: 16,602.0 ± 1057.0 N/s, post: 17,368.0 ± 1447.6 N/s, $p < 0.076$).

## Discussion

The aim of this study was to assess the neuromuscular and kinematics responses of the landing leg during single-leg drop landings before and after passive static loading of the knee joint capsule. Based upon previous research involving passive loading of the knee joint capsule, it was

**Table 4. Joint angular velocities.**

| | Maximum | | p-value | Minimum | | p-value |
|---|---|---|---|---|---|---|
| | Pre | Post | | Pre | Post | |
| Hip flexion | 204.9 (76.4) | 227.0 (46.5) | 0.43 | -60.9 (62.2) | -65.5 (73.5) | 0.78 |
| Knee flexion | 527.9 (131.6) | 544.1 (168.7) | 0.59 | -14.9 (25.2) | -27.9 (34.2) | **< 0.02** |
| Ankle flexion | 792.3 (104.9) | 544.1 (168.7) | 0.66 | -4.7 (21.3) | -17.0 (41.5) | 0.21 |
| Hip Abduction | 87.1 (56.9) | 56.9 (63.4) | **< 0.007** | -304.2 (101.2) | -291.7 (62.6) | 0.57 |
| Knee abduction | 90.0 (48.4) | 99.7 (50.5) | 0.53 | -528.5 (127.5)† | -399.9 (129.3) | **< 0.002** |
| Knee rotation | 306.3 (99.9) | 276.4 (89.5) | 0.30 | -125.9 (51.4) | -141.5 (45.2) | 0.46 |

Mean (sd) maximum and minimum angular velocities (˚/s) at the hip, knee, and ankle joints during pre- and post-loading landing conditions. $p < 0.05$ is indicated in bold.

**Table 5. Joint ranges of motion.**

| Angle (°) | Pre | Post | p-value |
|---|---|---|---|
| Hip flexion | 15.9 (8.6) | 16.6 (8.8) | 0.64 |
| Knee flexion | 46.3 (9.8) | 46.8 (9.8) | 0.73 |
| Ankle flexion | 56.2 (8.5) | 52.6 (8.5) | **< 0.006** |
| Hip abduction | 19.3 (11.1) | 18.5 (11.4) | 0.71 |
| Knee abduction | 21.5 (13.0) | 21.9 (12.8) | 0.82 |
| Knee rotation | 14.5 (8.1) | 14.2 (8.1) | 0.79 |

Mean (sd) values of ROM of the hip, knee, and ankle joints during pre- and post-loading landing conditions.
p < 0.05 is indicated in bold.

believed that neuromuscular and biomechanical behaviors would be modified in the lower extremity at landing. The reasoning for this study was twofold: 1) mechanical loading of the viscoelastic passive tissues is known to influence mechanical behavior changes of the affected (loaded) tissues, as well as the EMG response of the surrounding muscles, and 2) in the absence of neuromuscular fatigue it is not known how the lower extremity will respond to a perturbation given during a functional activity once passive loading of the knee joint is performed. The initial hypothesis predicted a reduction in EMG amplitude of the muscles surrounding the knee joint at landing after passive knee joint loading. However, this hypothesis was not supported based upon the results. There were no significant neuromuscular changes between pre- and post-loading landing conditions. It was also assumed that the neuromuscular system would compensate for the reduced mechanical tension within the passive connective tissues to increase coactivation in the drop phase prior to landing. This, however, was not substantiated in the data and cannot be considered a control mechanism of the leg at landing in this study. The second hypothesis regarding compensation of joint motion due to the passive

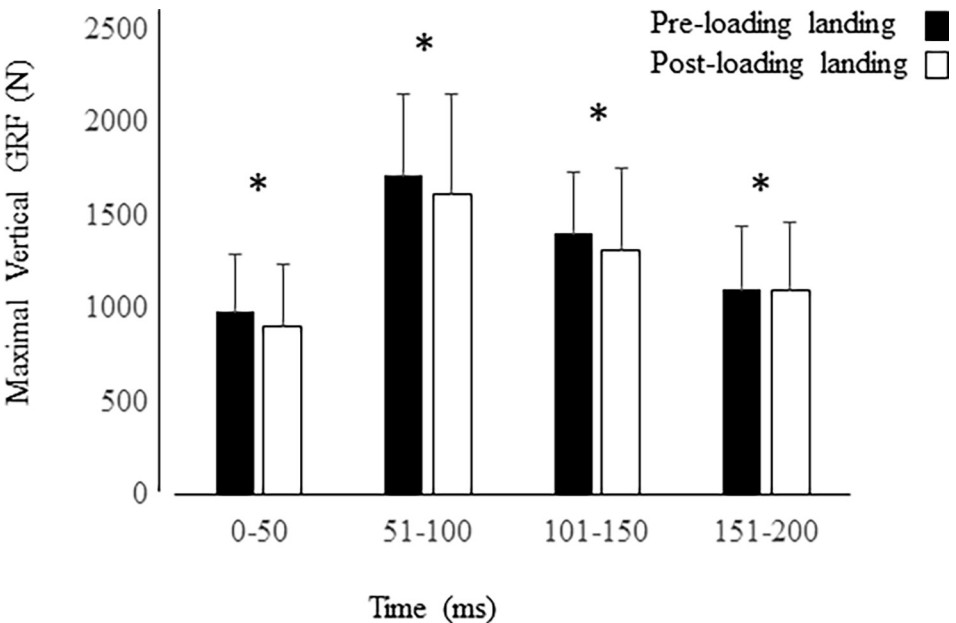

**Fig 4. Interval VGRF.** Mean (sd) maximal vertical ground reaction forces measured at 50 ms intervals from initial landing to 200ms during each condition (pre and post). * indicates significant difference between each 50ms time interval (p < 0.001).

loading at the knee joint capsule was partially supported. Distinct modifications to hip, knee, and ankle kinematics during landing resulted from the static loading. Further, the overall average VGRF decreased after static loading, while no change in the maximal force was evident.

## Electromyography

Neuromuscular control determined the response of the landing leg during the drop landing. Low level myoelectric activity from both hamstrings and quadriceps muscles groups assisted in preparing the leg for landing. This feed-forward mechanism allows the neuromuscular system to engage immediately once contact with the support surface is initiated to prevent the leg from buckling. Once landing occurred, greater muscle responses were observed to reduce joint angular motions. There were significant differences between the muscles groups at landing. Overall, maximum NEMG values were greatest in the RF group and least in the VM muscles group. However, average NEMG values were greatest from the VL group, and least in the VM group. This is expected as the VL muscles group has been reported to be activated at higher relative levels during similar tasks [22].

The descent phase of landing requires knee extensor muscles to perform eccentric actions to diminish flexion of the knee joint. The NEMG signals of all muscles examined increased as the knee approached maximum flexion. The contribution of the VL and RF muscles were much greater than that of the VM, indicating differential control within the knee extensors. This may indicate the inability of the VM muscle group to provide a primary role in joint stability, which may be due to architectural factors [29]. The increased myoelectric amplitudes are expected from the knee extensors, but not necessarily from the knee flexor muscles. The SM muscle group showed high activation levels in both conditions indicating that this muscle group was more actively involved in the control of leg mechanics compared to the BF muscle. There are two explanations which exemplify the activation of the hamstrings at landing: 1) the hamstring muscle activities are indicative of control at the hip joint to reduce hip joint flexion motion during the landing phase, and 2) sufficient hamstring muscle activity is required to compensate for anterior translation of the tibia [30].

Reduced BF muscle activity is suggested to increase knee internal rotation in a small sample of female athletes [31]. In patients who have undergone ACL reconstruction, modifying landing instruction to increase knee flexion at landing was reported to also reduce BF activity [27]. Although the BF activity was relatively lower than other muscles surrounding the knee, this did not influence overall knee flexion at landing (Table 3).

## Kinematics

Compensatory changes in the movement velocities at the knee and hip during the landing phase highlight the modified control the musculoskeletal system uses to respond to the dynamic loading. First, the rate of hip abduction at landing was significantly reduced (Table 4) even though the displacement of the hip joint during landing did not change between landing conditions (Table 3). Reduced hip abduction angular velocity suggests a potential increase in hip adductor contribution during the landing. This negative hip abduction velocity may compensate for the mechanics observed at the knee joint. The ability of the leg to absorb the shock at landing may have been due to greater emphasis of control at the knee as greater negative knee flexion angular velocity was observed. In addition, a reduced knee abduction negative velocity would indicate a greater control of frontal plane knee rotations and less mechanical energy being absorbed. Norcross et al. [32] reported knee landing kinematics differences between ACL injury risk groups and noted greater increased ligament loading with greater

energy absorption. It is possible that the mechanical energy absorbed by each joint at landing was modified and could explain how control at each joint was performed.

The knee joint kinematics may have influenced the range of motion observed from the ankle joint after the passive loading. Overall ankle joint range of motion decreased, while no maximum or minimum angular displacement measures were different between the landing conditions. Fong et al. [28] noted that passive ankle joint range of motion was related to greater knee flexion at landing leading to reduced stress in the ACL, specifically. Although greater range of motion can reduce the forces acting upon the joints at landing, Butler et al. [33] suggest increased joint stiffness is important for successful landing mechanics. However, increased limb stiffness is also a factor in potential lower extremity injury, particularly in female athletes [34,35]. Additionally, maximal hip abduction velocity decreased, minimum knee flexion angular velocity increased in magnitude, and knee abduction velocity decreased indicating potential neuromuscular control enhancement of the muscles surrounding the joints.

## Kinetics

Significant reductions in the average VGRF after static loading of the knee joint was present. Initially, it is possible that the landing style changed between conditions, however, there was no kinematics evidence to suggest foot position changed at the initial contact with the support surface. Although a reduced range of motion at the ankle joint in plantar-dorsiflexion was observed, this was not believed to influence the landing, especially within the first 10 ms of the landing, which is a critical time period of knee injuries at landing. Kernozek et al. [14] observed a non-significant trend of reduced maximal VGRF, as well as reduced internal joint reaction forces during drop landings performed after fatiguing the thigh musculature. Similarly, Laughlin et al. [36] reported reduced maximal VGRFs and maximal ACL forces when female participants were instructed to land with greater knee flexion during drop landing. They observed kinematics differences at initial contact and maximal ACL force from the hip and knee joints which explained their findings.

Although not tested, the stiffness of the leg influences the ability of the system to resist external loads applied. In particular, the musculotendinous stiffness influences the knee joint loading and ability to dissipate mechanical energy. Greater joint stiffness at landing when the knee is more extended leads to increased injury potential [37,38]. Hamstring musculotendinous stiffness has been reported to reduce the loading of the ACL and limit frontal plane rotations [39]. This is significant as the current study was implemented to reduce stiffness in the tissues within and surrounding the knee joint. In addition, joint stiffness has been reported to be greater in females compared to males at landing [35]. This may serve as an initial protective mechanism for the joint at landing, but may act to increase the chances of knee ligament injury.

## Knee joint loading

Isolation of the knee joint utilizing specific loading schemes to assess the neuromuscular responses of the surrounding joint musculature provides biomechanical information of the factors associated with knee joint injury mechanisms, in the absence of neuromuscular fatigue. When muscles become fatigued more of the load/stress is transferred to the passive viscoelastic tissues to maintain joint integrity during functional movements. Although not a functional loading scheme, the passive loading implemented in the current study has been shown to elicit creep behavior of the tissues within and surrounding the knee joint capsule [19,20]. Evidence of the influence of these mechanical creep experiments has provided mixed information, but this is also dependent upon the specific intentions of each study. Cheng et al. [19] initiated

posterior loading of the tibia to elicit posterior cruciate ligament creep and reported reduced co-activation of the antagonist thigh muscles during knee extension activities. Chu et al. [20], however, noted increased force and agonist activation during maximal effort knee extension exercises with no changes in antagonist (hamstring) activities. Further evidence of passive tissue loading within and surrounding the knee joint demonstrates reduced agonist and antagonist muscle activities in maximal efforts [21,26], indicating a potential neuromuscular inhibition which may impede function of the muscle during activity. Specifically, Nuccio et al. [21] report significant reductions in the biceps femoris muscle activity after cyclic loading during both knee flexion and knee extension static efforts.

It must be emphasized that isolated loading of specific tissues, such as the ACL or PCL in the knee joint capsule, are not directly linked in *in vivo* studies. Unlike animal models where tissues can be isolated for perturbation/loading to determine the effects of mechanical manipulation of the specimen [40,41,42], there are factors which constitute how human models can be interpreted. Loading of the knee joint involves applying mechanical creep to the surrounding musculotendonous units, ligamentous tissues, meniscus, and other connective tissues which assist in maintaining the functional dynamics of the knee during physical activity. Particularly when applying these anteriorly directed loads, the musculotendonous units of the hamstrings muscle group can be strained leading to potential modifications in the activation level and stiffness of the muscle [40]. Shear stress of the meniscus during anterior loading is reported to differ between femoral and tibial anterior and posterior attachments, as well as medial and lateral attachments leading to an overall disparity in load distribution in ACL-deficient knees [43].

### Limitations

There were limitations to this study which need to be addressed. Relatively moderate loads were applied to the knee joints. This was performed to elicit a creep behavior in the viscoelastic connective tissues, as shown by Chu et al.[20]. Increased loads applied to the knee joints may increase the creep response and this may modify the results presented in this paper to coincide with loads incurred during athletic events. The dominant leg was assessed in the current study. Injuries occur in both dominant and non-dominant leg, and the responses to similar loads may be different between these limbs [44,45] Therefore, additional measures are required to assess both knees in future studies. Further, sex-specific differences between men and women need to be examined to better understand how these loading schemes influence neuromechanical responses.

### Conclusions

Implementation of a static load to the knee joint capsule modified movement parameters during a drop landing performance. Tissue-level behavioral changes may be present to influence how the lower extremity joints respond to dynamic loading. Neuromuscular modifications were not present between the landing conditions indicating that this loading scheme does not result in altered neuromuscular control. Additional research is warranted to examine potential modifications to the loading schemes to further understand how the neuromechanics of the lower extremities are modified when controlling for fatigue.

### Supporting information

**S1 Table. Mean (sd) average EMG at pre- and post-loading landing at 200 and 100ms prior to landing, and at 50 ms intervals at landing.**
(DOCX)

**S2 Table. Mean (sd) maximal EMG from each muscle group during pre- and post-loading landing conditions.**
(DOCX)

**S3 Table. Mean (sd) maximal VGRFs at each 50ms interval of pre- and post-loading landing up to 200 ms.**
(DOCX)

## Author Contributions

**Conceptualization:** Michael W. Olson.

**Data curation:** Michael W. Olson.

**Formal analysis:** Michael W. Olson.

**Investigation:** Michael W. Olson.

**Methodology:** Michael W. Olson.

**Project administration:** Michael W. Olson.

**Resources:** Michael W. Olson.

**Writing – original draft:** Michael W. Olson.

**Writing – review & editing:** Michael W. Olson.

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
