## [Decision Letter · Decision Letter 0]

19 Sep 2019

PONE-D-19-17819

STATIC LOADING OF THE KNEE JOINT RESULTS IN MODIFIED SINGLE LEG LANDING BIOMECHANICS

PLOS ONE

Dear Dr. Olson,

Thank you for submitting your manuscript to PLOS ONE. After careful consideration, we feel that it has merit but does not fully meet PLOS ONE’s publication criteria as it currently stands. Therefore, we invite you to submit a revised version of the manuscript that addresses the points raised during the review process.

We would appreciate receiving your revised manuscript by Nov 03 2019 11:59PM. To enhance the reproducibility of your results, we recommend that if applicable you deposit your laboratory protocols in protocols.io, where a protocol can be assigned its own identifier (DOI) such that it can be cited independently in the future. For instructions see: http://journals.plos.org/plosone/s/submission-guidelines#loc-laboratory-protocols

We look forward to receiving your revised manuscript.

Kind regards,

Dragan Mirkov, Ph.D.

Academic Editor

PLOS ONE

Journal Requirements:

2. Thank you for including your ethics statement: "Southern Illinois University Carbondale Human Subject Committee

Approval # 15277

written consent was provided by each participant"

a. Please amend your current ethics statement to confirm that your named institutional review board or ethics committee specifically approved this study.

Reviewers' comments:

Reviewer's Responses to Questions

**Comments to the Author**

1. Is the manuscript technically sound, and do the data support the conclusions?

Reviewer #1: Partly

Reviewer #2: Partly

2. Has the statistical analysis been performed appropriately and rigorously? 

Reviewer #1: Yes

Reviewer #2: No

3. Have the authors made all data underlying the findings in their manuscript fully available?

Reviewer #1: Yes

Reviewer #2: No

4. Is the manuscript presented in an intelligible fashion and written in standard English?

Reviewer #1: Yes

Reviewer #2: Yes

5. Review Comments to the Author

Reviewer #1: Research is conducted according the paper guidelines. Methodology and project design are set correctly. The only objection is the small number of subjects who participated in the study. It is not possible to make a quality conclusion based on such a number. It would be necessary to increase the number and adjust the statistical analysis in order to reach a good conclusion.

Reviewer #2: Review the manuscript titled “Static loading of the knee joint results in modified single leg landing biomechanics”

Manuscript Number: PONE-D-19-17819

The aim of this study was to investigate the muscular, kinetic and kinematic response of the lower limb during landing following anterior passive loading of the capsuloligamentous structures of the knee joint. The experimental setup used in the present study was similar to the one used by Chu et al (2003) who also investigated the effect of creep that developed in the anterior cruciate ligament due to prolonged static loading on the reflexive activation of the associated musculature.

Even though the scientific value of this study is unquestionable, as it enlightens the mechanisms that underlie knee injuries, there are some issues that require author’s respond, particularly with regard to the method, the statistical analysis of the research data as well as the presentation and discussion of the results, before the study is considered for publication.

General comments

Method

• The authors should justify the choice of loads that they implemented for static loading of the knee joint as these could be considered light to mild in comparison to the loads that are developed in sporting activities such as during landing. This issue should be discussed more thoroughly and possibly introduced as one of the limitations of the present study.

• The authors should also justify their choice to use the dominant leg (the one the participants use for kicking a ball) in their study as previous studies have demonstrated significant differences between the dominant and non-dominant leg regarding knee joint stability during unilateral landing (Ludwig et al 2017; Herrington 2011).

• Given the complexity of the experiment, it is suggested that the researchers provide photographic material for the experimental set up and procedure.

Statistical analysis

The general sense that someone gets from reading this manuscript is data over-analysis. The authors performed several statistical tests in order to identify potential effect of static loading of the knee’s passive structures during landing. However, there is a mismatch (in my opinion) between the statistical tests that were performed, the presentation of the Results and the data listed or depicted in Tables and Graphs, respectively. More specifically:

• The authors performed a Two-Way (condition x muscle) ANOVA to compare average and maximum normalized EMG of each muscle involved during landing between pre and post loading conditions (the results are listed in Table 1); they also performed a 5 (muscles) x 2 (conditions) x 4 (time intervals) Two-Way ANOVA for average normalized EMG (the results are graphically presented in Figure 1). It is clear that in the latter case the authors attempted to investigate how muscles’ EMG behave over time during landing; however in Figure 1 the normalized EMG that was recorded before landing (200-0 and 100-0 ms) is also depicted. Why the authors did not include the pre-landing EMG’s in the statistical analysis? Their decision should be is justified and, in any case, they should consider modifying Figure 1, as in its current state it seems that both pre-landing and during landing EMG responses were also compared.

• Similarly, EMG muscles ratios have been compare with two different statistical tests. A muscle ratio x condition Two-Way ANOVA was used to compare average and maximum EMG ratios between the pre- and post-loading condition (the results are listed in Table 3 and 4) and again a 6 (muscle ratios) x 2 (conditions) x 5 (time intervals) Two-Way ANOVA for muscles EMG ratios (results are graphically presented in Figure 2). In this case (i) the authors did not report whether muscles EMG ratios were based on the average or the maximum EMG activity, (ii) they included 5 time intervals whilst the time intervals during landing were 4; which was the 5th time interval? Did the authors include in their analysis a pre-landing time interval (200-0 or 100-0 ms)? And if they do so, why they did not the same for average and maximum EMG activity of the muscles under investigation? (iii) an additional column was added in Figure 2 (the one that represents total EMG ratio for each time interval) but it should be clarified whether or not it was included in the statistical analysis. The authors should consider omitting the “total EMG ratio column” particularly if its scientific value is meaningless as it confuses the reader (see next comment).

• In both aforementioned cases the “overall” average and maximum EMG as well as the “overall” muscle EMG ratios is presented in Tables 2-4. In Figure 2 and Figure 3 a “total” EMG ratio column and a “combined” vertical ground reaction forces column (VGRF), respectively is also presented. The authors should justify whether (i) the “overall”, “total” or “combined” are terms that were used interchangeably, (ii) these terms represent data obtained at the pre-landing and during landing condition (which in my opinion make sense) or data obtained before loading and after loading of the passive knee structures. The legends in Tables 3 and 4 add more confusion to the reader as it is not clear whether the data listed in Tables represent “… average and maximal muscle ratios during landing phase between pre and post landing conditions” or “… loading conditions”. In any case the authors should consider omitting the “overall”, “total” or “combined” data, as in my opinion this information neither adds more scientific value to this research nor it is thoroughly discussed.

• For VGRF the authors (i) reported the test that they used in statistical analysis, (ii) in the Results section they reported no statistical differences between conditions with regard to absolute peak and normalized peak forces but (iii) in the Discussion section they stated that “significant reductions in the average VGRF after static loading of the knee was present” (Page 20, Lines 429-430). This information was probably extracted from the graph in Figure 3 (?), but the authors neglected to report the level of statistical significances either in the text or in the graph.

• The statistical tests used for maximum and minimum angular displacement, angular velocities and ROM comparisons between conditions were not reported.

Discussion

The authors should discuss the limitations of the study and propose future research based on the findings and / or the methodological Inadequacies of the present study.

Minor comments

Page 3, Line 47: The authors should consider replacing “… stress and strains which …” with “… stress and strains, which…”

Page 3, Line 61: The authors should clarify whether they are referring to the anatomical (malalignment of mechanical axes) or dynamic “… knee valgus …”

Page 4, Line 68: The authors should consider replacing “…musculotendonous…” with “…musculotendinous…”

Page 6, Line 126: The authors should describe the technique used, if any, for identification of motor points and/or cite related references.

Page 7, Lines 148-152: The authors should clarify the number of test trials that were actually included in data analysis as according to their protocol 10 test trials were required but eventually “… 5 sufficient trials…” were enough. Does this mean that the average of 5 trials out of the 10 was included in the data analysis and, if yes, which trials were discarded if more than 5 trials were recorded?

Page 13, Line 276: The authors should replace (or not) the “… pre and post landing conditions” with “… pre and post loading conditions” in the legend of Table 3. The same applies for the legend of Table 4 (Page 14).

Page 13, Table 3: The authors should consider omitting the “overall” data (unless they explain and justify its use) since, in my opinion, it is confusing, not related with the initial hypothesis and eventually not discussed, although it appears that is the only data sets that present statistical significances. The same applies for the legend of Table 4 (Page 14).

Page 15, Line 307: The authors should consider replacing “… (p>0.071)” with “… (p<0.071)”

Page 19, Lines 400-401: The authors should consider replacing “… overall knee rotation at landing” with “… overall knee flexion at landing”. They should also consider modifying Table 5 as in this Table the minimum and maximum, but no “overall”, displacement is listed.

Page 19, Line 407: The authors should consider replacing “… (Table 4)” should with “… (Table 5)” as the latter one display data values of joints angular displacements while the former displays data values of EMG muscle ratios.

Page 19, Lines 410-411: The authors should specify the negative values in both the text and in Tables 5 and 6.

Figure 1: The units for NEMG should be added in the vertical axis

Figure 1-3: The author should use the same title in the horizontal axes (Time or Time intervals) in all Figures. The levels of statistical significance may also be indicated in a revised version of the manuscript provided that the proposed amendments have been taken into account.

Figure 3: The legend should be modified according to the other graphs for consistency. For example: Pre-loading � instead of Pre - black

6. PLOS authors have the option to publish the peer review history of their article (what does this mean?). If published, this will include your full peer review and any attached files.

Reviewer #1: No

Reviewer #2: Yes: Dimitris Mandalidis, Assistant Professor, Department of Physical Education and Sports Science, National and Kapodistrian University of Athens

---

## [Author Response · Author response to Decision Letter 0]

14 Nov 2019

Dear Dr. Mirkov,

The suggestions and comments posed by you are the reviewers are greatly appreciated. After reviewing the feedback, the manuscript was revised to better communicate the intensions of the study. A manuscript with the track changes, to show the edits made, and a clean copy of the revised manuscript have been provided. Reponses to the reviewers have been indicated with R, and in bold type.

Thank you.

Response to Reviewers

Journal Requirements:

R: Thank you for this comment. The manuscript has been formatted to the specifications of the journal.

2. Thank you for including your ethics statement: "Southern Illinois University Carbondale Human Subject Committee

Approval # 15277

written consent was provided by each participant"

a. Please amend your current ethics statement to confirm that your named institutional review board or ethics committee specifically approved this study.

R: The ethical statement has been amended and added as requested (p.5. lines 106-108).

Reviewers' comments:

Reviewer #1: Research is conducted according the paper guidelines. Methodology and project design are set correctly. The only objection is the small number of subjects who participated in the study. It is not possible to make a quality conclusion based on such a number. It would be necessary to increase the number and adjust the statistical analysis in order to reach a good conclusion.

R: Thank you for highlighting this item. It is believed that a quality conclusion can be established, based upon the reported data. The sample size was believed to be sufficient for this study. Areas of analysis have been streamlined to better present the information. As far as the statistical analysis is concerned, the explanation of the statistical procedures have been revised to more accurately describe how the data were assessed (p.11, lines 225-238).

Reviewer #2: Review the manuscript titled “Static loading of the knee joint results in modified single leg landing biomechanics”

Manuscript Number: PONE-D-19-17819

The aim of this study was to investigate the muscular, kinetic and kinematic response of the lower limb during landing following anterior passive loading of the capsuloligamentous structures of the knee joint. The experimental setup used in the present study was similar to the one used by Chu et al (2003) who also investigated the effect of creep that developed in the anterior cruciate ligament due to prolonged static loading on the reflexive activation of the associated musculature.

Even though the scientific value of this study is unquestionable, as it enlightens the mechanisms that underlie knee injuries, there are some issues that require author’s respond, particularly with regard to the method, the statistical analysis of the research data as well as the presentation and discussion of the results, before the study is considered for publication.

General comments

Method

• The authors should justify the choice of loads that they implemented for static loading of the knee joint as these could be considered light to mild in comparison to the loads that are developed in sporting activities such as during landing. This issue should be discussed more thoroughly and possibly introduced as one of the limitations of the present study.

R: This is a very good issue presented by the reviewer. The load type was performed to replicate the study performed by Chu et al. (2003). However, as the reviewer suggests, these loads may not have been sufficient for some participants. This has been added as a limitation of the study (p. 22 lines 452-456).

• The authors should also justify their choice to use the dominant leg (the one the participants use for kicking a ball) in their study as previous studies have demonstrated significant differences between the dominant and non-dominant leg regarding knee joint stability during unilateral landing (Ludwig et al 2017; Herrington 2011).

R: The use of the dominant leg has been used throughout the literature (Heebner et al., 2017; Jenkins et al., 2017; Kernozek et al., 2008). However, this does not discount the reviewer’s suggestion. This has been added as a limitation and framed as a topic for future work (p.22, lines 456-459).

• Given the complexity of the experiment, it is suggested that the researchers provide photographic material for the experimental set up and procedure.

R: Additional figure have been added to illustrate the experimental loading and landing protocols (p. 8 and 9). 

Statistical analysis

The general sense that someone gets from reading this manuscript is data over-analysis. The authors performed several statistical tests in order to identify potential effect of static loading of the knee’s passive structures during landing. However, there is a mismatch (in my opinion) between the statistical tests that were performed, the presentation of the Results and the data listed or depicted in Tables and Graphs, respectively. More specifically:

• The authors performed a Two-Way (condition x muscle) ANOVA to compare average and maximum normalized EMG of each muscle involved during landing between pre and post loading conditions (the results are listed in Table 1); they also performed a 5 (muscles) x 2 (conditions) x 4 (time intervals) Two-Way ANOVA for average normalized EMG (the results are graphically presented in Figure 1). It is clear that in the latter case the authors attempted to investigate how muscles’ EMG behave over time during landing; however in Figure 1 the normalized EMG that was recorded before landing (200-0 and 100-0 ms) is also depicted. Why the authors did not include the pre-landing EMG’s in the statistical analysis? Their decision should be is justified and, in any case, they should consider modifying Figure 1, as in its current state it seems that both pre-landing and during landing EMG responses were also compared.

R: Thank you for finding these issues in the statistical analysis section. For clarity, the explanation of the data analysis for each dependent variable has been revised. When EMG analysis is conducted, the 200 ms before and 200 ms after landing are explicitly stated.

• Similarly, EMG muscles ratios have been compare with two different statistical tests. A muscle ratio x condition Two-Way ANOVA was used to compare average and maximum EMG ratios between the pre- and post-loading condition (the results are listed in Table 3 and 4) and again a 6 (muscle ratios) x 2 (conditions) x 5 (time intervals) Two-Way ANOVA for muscles EMG ratios (results are graphically presented in Figure 2). In this case (i) the authors did not report whether muscles EMG ratios were based on the average or the maximum EMG activity, (ii) they included 5 time intervals whilst the time intervals during landing were 4; which was the 5th time interval? Did the authors include in their analysis a pre-landing time interval (200-0 or 100-0 ms)? And if they do so, why they did not the same for average and maximum EMG activity of the muscles under investigation? (iii) an additional column was added in Figure 2 (the one that represents total EMG ratio for each time interval) but it should be clarified whether or not it was included in the statistical analysis. The authors should consider omitting the “total EMG ratio column” particularly if its scientific value is meaningless as it confuses the reader (see next comment).

R: To clarify the data and simplify the Results section, the ratio data have now been excluded from the data analysis. 

• In both aforementioned cases the “overall” average and maximum EMG as well as the “overall” muscle EMG ratios is presented in Tables 2-4. In Figure 2 and Figure 3 a “total” EMG ratio column and a “combined” vertical ground reaction forces column (VGRF), respectively is also presented. The authors should justify whether (i) the “overall”, “total” or “combined” are terms that were used interchangeably, (ii) these terms represent data obtained at the pre-landing and during landing condition (which in my opinion make sense) or data obtained before loading and after loading of the passive knee structures. The legends in Tables 3 and 4 add more confusion to the reader as it is not clear whether the data listed in Tables represent “… average and maximal muscle ratios during landing phase between pre and post landing conditions” or “… loading conditions”. In any case the authors should consider omitting the “overall”, “total” or “combined” data, as in my opinion this information neither adds more scientific value to this research nor it is thoroughly discussed.

R: There is agreement with the reviewer that the “overall” and “total” data do not provide meaningful information. These have been removed from the figures and tables. The Figure legends and table legends have been updated to reflect a clearer reflection of the presented data.

• For VGRF the authors (i) reported the test that they used in statistical analysis, (ii) in the Results section they reported no statistical differences between conditions with regard to absolute peak and normalized peak forces but (iii) in the Discussion section they stated that “significant reductions in the average VGRF after static loading of the knee was present” (Page 20, Lines 429-430). This information was probably extracted from the graph in Figure 3 (?), but the authors neglected to report the level of statistical significances either in the text or in the graph.

R: These VGRF results have been included in the Results section of the revised manuscript (p. 17, lines 308-310). Additionally, the normalized forces have been omitted from the revised manuscript. 

• The statistical tests used for maximum and minimum angular displacement, angular velocities and ROM comparisons between conditions were not reported.

R: These statistical tests are now included in the revised manuscript (p. 11, lines 225-227).

Discussion

The authors should discuss the limitations of the study and propose future research based on the findings and / or the methodological Inadequacies of the present study.

R: A section has been included to highlight the limitations and potential future studies related to these limitations (p. 22, lines 452-461).

Minor comments

Page 3, Line 47: The authors should consider replacing “… stress and strains which …” with “… stress and strains, which…”

R: This has been changed, as suggested by the reviewer.

Page 3, Line 61: The authors should clarify whether they are referring to the anatomical (malalignment of mechanical axes) or dynamic “… knee valgus …”

R: This point has been clarified to the dynamics of the movement.

Page 4, Line 68: The authors should consider replacing “…musculotendonous…” with “…musculotendinous…”

R: This spelling has been corrected throughout the document.

Page 6, Line 126: The authors should describe the technique used, if any, for identification of motor points and/or cite related references.

R: References for electrode placement have been provided in the methods (p. 6-7, lines 127-131).

Page 7, Lines 148-152: The authors should clarify the number of test trials that were actually included in data analysis as according to their protocol 10 test trials were required but eventually “… 5 sufficient trials…” were enough. Does this mean that the average of 5 trials out of the 10 was included in the data analysis and, if yes, which trials were discarded if more than 5 trials were recorded?

R: This has been clarified in the revised document.

Page 13, Line 276: The authors should replace (or not) the “… pre and post landing conditions” with “… pre and post loading conditions” in the legend of Table 3. The same applies for the legend of Table 4 (Page 14).

R: These changes have been made in the revised manuscript.

Page 13, Table 3: The authors should consider omitting the “overall” data (unless they explain and justify its use) since, in my opinion, it is confusing, not related with the initial hypothesis and eventually not discussed, although it appears that is the only data sets that present statistical significances. The same applies for the legend of Table 4 (Page 14).

R: As stated previously, this has been deleted from the figures, tables, and text of the manuscript.

Page 15, Line 307: The authors should consider replacing “… (p>0.071)” with “… (p<0.071)”

R: This has been changed.

Page 19, Lines 400-401: The authors should consider replacing “… overall knee rotation at landing” with “… overall knee flexion at landing”. They should also consider modifying Table 5 as in this Table the minimum and maximum, but no “overall”, displacement is listed.

R: These changes have been made to the manuscript.

Page 19, Line 407: The authors should consider replacing “… (Table 4)” should with “… (Table 5)” as the latter one display data values of joints angular displacements while the former displays data values of EMG muscle ratios.

R: This has been corrected in the manuscript.

Page 19, Lines 410-411: The authors should specify the negative values in both the text and in Tables 5 and 6.

R: This has been updated in the revised manuscript.

Figure 1: The units for NEMG should be added in the vertical axis

R: There are no units of normalized EMG. Rather, the data are provided with reference to the maximal EMG value of the respective muscle.

Figure 1-3: The author should use the same title in the horizontal axes (Time or Time intervals) in all Figures. The levels of statistical significance may also be indicated in a revised version of the manuscript provided that the proposed amendments have been taken into account.

Figure 3: The legend should be modified according to the other graphs for consistency. For example: Pre-loading � instead of Pre – black

 R: These changes have been made in the revised manuscript.

---

## [Decision Letter · Decision Letter 1]

18 Dec 2019

PONE-D-19-17819R1

Static loading of the knee joint results in modified single leg landing biomechanics

PLOS ONE

Dear Dr. Olson,

Thank you for submitting your manuscript to PLOS ONE. After careful consideration, we feel that it has merit but does not fully meet PLOS ONE’s publication criteria as it currently stands. Therefore, we invite you to submit a revised version of the manuscript that addresses the points raised during the review process.

We would appreciate receiving your revised manuscript by Feb 01 2020 11:59PM. To enhance the reproducibility of your results, we recommend that if applicable you deposit your laboratory protocols in protocols.io, where a protocol can be assigned its own identifier (DOI) such that it can be cited independently in the future. For instructions see: http://journals.plos.org/plosone/s/submission-guidelines#loc-laboratory-protocols

We look forward to receiving your revised manuscript.

Kind regards,

Dragan Mirkov, Ph.D.

Academic Editor

PLOS ONE

Reviewers' comments:

Reviewer's Responses to Questions

**Comments to the Author**

1. If the authors have adequately addressed your comments raised in a previous round of review and you feel that this manuscript is now acceptable for publication, you may indicate that here to bypass the “Comments to the Author” section, enter your conflict of interest statement in the “Confidential to Editor” section, and submit your "Accept" recommendation.

Reviewer #1: All comments have been addressed

Reviewer #2: All comments have been addressed

2. Is the manuscript technically sound, and do the data support the conclusions?

Reviewer #1: Yes

Reviewer #2: Yes

3. Has the statistical analysis been performed appropriately and rigorously? 

Reviewer #1: Yes

Reviewer #2: No

4. Have the authors made all data underlying the findings in their manuscript fully available?

Reviewer #1: Yes

Reviewer #2: No

5. Is the manuscript presented in an intelligible fashion and written in standard English?

Reviewer #1: Yes

Reviewer #2: Yes

6. Review Comments to the Author

Reviewer #1: I still think that such a quality and correct methodology should have more subjects in the statistical analysis.

Reviewer #2: Review of the manuscript titled “Static loading of the knee joint results in modified single leg landing biomechanics”. Manuscript Number: PONE-D-19-17819.R1

The investigator of this study responded to all the comments and queries that were highlighted after reviewing the first version of the manuscript. However, there are several points that should be clarified, particularly with regard to the statistical analysis of the research data as well as the presentation and discussion of the results, before the study is considered for publication.

Abstract

Page 2, Lines 38-39: The authors should clarify the abbreviations aVGRF and NAEMG as they do not appear in the text

Methods

Pages 10-11, Lines 215-219: The author should consider re-writing the sentence following the suggestions presented below.

Statistical analysis

Page 11, Lines 224-234: The author should (i) clarify why Pre- vs. Post- comparisons, for angular displacement and velocity variables as well as for average and maximal forces, RFD and VGRF, are performed with a One-Way Analysis Of Variance and not a paired-t test, (ii) report the statistical test that he used for the analysis of the Force Timing-related data, (iii) consider to refer to the statistical tests used for each variable in the same order that it is presented in the Results section and discussed in the Discussion section. Alternatively, the statistical tests could be presented in groups (e.g. first the statistical test used for paired comparisons and then the ANOVA tests).

Results

Pages 12-16: The author should specify what the “pre- and post- landing condition” and the “pre- and post-loading condition” are. It appears that (i) the terms “pre- and post- landing” are used when the comparisons are intended to assess the effect of the pre-landing-induced feed forward mechanism on certain variables (e.g. muscle activity) and (ii) the terms “pre- and post-loading” when the comparisons aimed to assess the effect of passive loading of the knee on a particular variable.

An example: Table 2 presents comparisons between pre- and post-landing conditions for “Overall Average and Maximum NEMG” data. This is okay since the author stated (Pages 10-11, Lines 215-219) that such comparisons will provide information on the effect of the feed forward mechanism. However, it is not clear whether this represents the data that recorded before (pre-) or after (post-) passive loading of the knee. The author should (i) consider presenting pre- and post-loading and pre- and post-landing data in Tables or Figures as he did for the average NEMG data (Figure 3) and (ii) perform Two-Way ANOVAs for the maximum NEMG, as well as for all the other variables, as he did for average NEMG and VGRF.

Such clarifications should be made in the following cases, but generally this should be done whenever is deemed necessary.

Page 12, Line 250: In the text

Page 13, Line 253 and 255: In the text and the legend of Table 2. It should be clarified whether the data presented refers to the pre-loading or the post-loading condition

Page 13, Line 257: Consider replacing “…during drop landing conditions.” with “…during drop landing.”

Page 13, Line 263-272: The author should (i) clarify in the text the "conditions" that he is referred to and (ii) report in the legend of Figure 3 that the data is referred to both pre- and post-loading conditions.

Page 14, Line 275 and 279: In the text and legend of Table 3 for angular displacement data

Page 14, Line 282-286 and Page 15, Line 294: In the text and legend of Table 4 for angular velocity data

Page 15, Line 296-297 and 303-304: In the text and legend of Table 5 for joint ROM data

Page 15-16, Lines 306-310: In the text and legend (Page 16, Line 312-314) of Figure 4 for VGRF (How is it possible to record VGRF before landing?).

Page 16, Line 316: The author should consider defining in the Method section the Force Timing variable and report the statistical test that it was used for its analysis before presenting the related Results, as this variable appears for the first time in this section. Furthermore, Figure 4 presents maximum VGRF and not Force Timing data the values of which remain unspecified.

Page 12, Line 252: The author should consider replacing “Maximal” with “Maximum” and “EMG” with “NEMG” wherever is deemed appropriate.

Table 2 and Figure 3: The presentation of the values in Table 2 and Figure 3 for Average NEMG is inconsistent. The values presented in Table 2 are percentages of MVIE whilst the values presented in Figure 3 vary between 0-1.5 but remain unspecified. The authors should consider revising either the data in the Table or the data in the Figure.

Tables 3-5: The author should clarify the movement of ankle - dorsiflexion or plantarflexion - that is presented in the Tables and to explain the negative numbers.

Figure 4: The author should consider replacing the labels “Pre-landing” and “Post-landing” with “Pre-loading” and “Post-loading” that is presented within Figure 4.

Discussion

The discussion is very well written but the author must explain which conditions were compared. This is particularly important for the understanding of the discussion related to the Kinematic and Kinetic data.

7. PLOS authors have the option to publish the peer review history of their article (what does this mean?). If published, this will include your full peer review and any attached files.

Reviewer #1: No

Reviewer #2: Yes: Dimitris Mandalidis, Assistant Professor, School of Physical Education and Sports Science, National and Kapodistrian University of Athens

---

## [Author Response · Author response to Decision Letter 1]

15 Jan 2020

Abstract

Page 2, Lines 38-39: The authors should clarify the abbreviations aVGRF and NAEMG as they do not appear in the text 

R: The abbreviations have been modified to agree with the abbreviations in the text.

Methods

Pages 10-11, Lines 215-219: The author should consider re-writing the sentence following the suggestions presented below.

R: please see the response after the following reviewer suggestion

Statistical analysis

Page 11, Lines 224-234: The author should (i) clarify why Pre- vs. Post- comparisons, for angular displacement and velocity variables as well as for average and maximal forces, RFD and VGRF, are performed with a One-Way Analysis Of Variance and not a paired-t test, (ii) report the statistical test that he used for the analysis of the Force Timing-related data, (iii) consider to refer to the statistical tests used for each variable in the same order that it is presented in the Results section and discussed in the Discussion section. Alternatively, the statistical tests could be presented in groups (e.g. first the statistical test used for paired comparisons and then the ANOVA tests).

R: It was believed that the ANOVA test was a valid statistical assessment of the values provided between the two conditions. In viewing the one-way ANOVA and paired-t-test options, both would provide the same outcome, thus the one-way ANOVA has been reported.

Results 

Pages 12-16: The author should specify what the “pre- and post- landing condition” and the “pre- and post-loading condition” are. It appears that (i) the terms “pre- and post- landing” are used when the comparisons are intended to assess the effect of the pre-landing-induced feed forward mechanism on certain variables (e.g. muscle activity) and (ii) the terms “pre- and post-loading” when the comparisons aimed to assess the effect of passive loading of the knee on a particular variable.

R: The text has been edited to clarify pre-loading landing and post-loading landing conditions – these are the only “conditions”, as defined in the Methods section

An example: Table 2 presents comparisons between pre- and post-landing conditions for “Overall Average and Maximum NEMG” data. This is okay since the author stated (Pages 10-11, Lines 215-219) that such comparisons will provide information on the effect of the feed forward mechanism. However, it is not clear whether this represents the data that recorded before (pre-) or after (post-) passive loading of the knee. The author should (i) consider presenting pre- and post-loading and pre- and post-landing data in Tables or Figures as he did for the average NEMG data (Figure 3) and (ii) perform Two-Way ANOVAs for the maximum NEMG, as well as for all the other variables, as he did for average NEMG and VGRF. 

Such clarifications should be made in the following cases, but generally this should be done whenever is deemed necessary.

R: All comparisons are between pre-loading landing and post-loading landing conditions. This nomenclature has been modified to reflect the clarity of the conditions in the text. The NEMG was evaluated 200 ms pre-landing and 200 ms post-landing, while the VGRF data were evaluated 200 ms post-landing in each condition. The data are presented in either figures or tables in the manuscript. The “overall average and maximal” values were removed for the first revision, so it is unclear what the reviewer is viewing.

Page 12, Line 250: In the text

R: the conditions have been defined in the Methods section for clarity

Page 13, Line 253 and 255: In the text and the legend of Table 2. It should be clarified whether the data presented refers to the pre-loading or the post-loading condition 

R: this has been clarified

Page 13, Line 257: Consider replacing “…during drop landing conditions.” with “…during drop landing.”

R: this has been clarified

Page 13, Line 263-272: The author should (i) clarify in the text the "conditions" that he is referred to and (ii) report in the legend of Figure 3 that the data is referred to both pre- and post-loading conditions.

R: these have been clarified

Page 14, Line 275 and 279: In the text and legend of Table 3 for angular displacement data

R: this has been clarified

Page 14, Line 282-286 and Page 15, Line 294: In the text and legend of Table 4 for angular velocity data

R: this has been clarified

Page 15, Line 296-297 and 303-304: In the text and legend of Table 5 for joint ROM data

R: this has been clarified

Page 15-16, Lines 306-310: In the text and legend (Page 16, Line 312-314) of Figure 4 for VGRF (How is it possible to record VGRF before landing?).

R: it is not. It is not clear what the reviewer is reading, but there has been no indication in this manuscript that forces were recorded before landing.

Page 16, Line 316: The author should consider defining in the Method section the Force Timing variable and report the statistical test that it was used for its analysis before presenting the related Results, as this variable appears for the first time in this section. Furthermore, Figure 4 presents maximum VGRF and not Force Timing data the values of which remain unspecified. 

R: this has been defined in the Methods section: “A 2-way ANOVA (condition x time interval) was used to assess average VGRF data during the landing phase, while one-way ANOVAs were used to compare maximal VGRF values between conditions”, p.11, lines 231-234. The caption of the section has been changed to clarify this variable assessment.

Page 12, Line 252: The author should consider replacing “Maximal” with “Maximum” and “EMG” with “NEMG” wherever is deemed appropriate.

R: this has been clarified

Table 2 and Figure 3: The presentation of the values in Table 2 and Figure 3 for Average NEMG is inconsistent. The values presented in Table 2 are percentages of MVIE whilst the values presented in Figure 3 vary between 0-1.5 but remain unspecified. The authors should consider revising either the data in the Table or the data in the Figure.

R: this has been clarified in Table 2

Tables 3-5: The author should clarify the movement of ankle - dorsiflexion or plantarflexion - that is presented in the Tables and to explain the negative numbers.

R: this has been clarified in the footnote for tables 3. Table 4 provides angular velocity data and the signs are indicative of positive and negative rates of movement, respectively. Table 5 represents ROM and does not require and explanation for the ankle

Figure 4: The author should consider replacing the labels “Pre-landing” and “Post-landing” with “Pre-loading” and “Post-loading” that is presented within Figure 4.

R: this has been modified.

Discussion 

The discussion is very well written but the author must explain which conditions were compared. This is particularly important for the understanding of the discussion related to the Kinematic and Kinetic data.

R: the conditions have been clarified in the Discussion section

---

## [Decision Letter · Decision Letter 2]

5 Feb 2020

Static loading of the knee joint results in modified single leg landing biomechanics

PONE-D-19-17819R2

Dear Dr. Olson,

We are pleased to inform you that your manuscript has been judged scientifically suitable for publication and will be formally accepted for publication once it complies with all outstanding technical requirements.

With kind regards,

Dragan Mirkov, Ph.D.

Academic Editor

PLOS ONE

Additional Editor Comments (optional):

Reviewers' comments:

Reviewer's Responses to Questions

**Comments to the Author**

1. If the authors have adequately addressed your comments raised in a previous round of review and you feel that this manuscript is now acceptable for publication, you may indicate that here to bypass the “Comments to the Author” section, enter your conflict of interest statement in the “Confidential to Editor” section, and submit your "Accept" recommendation.

Reviewer #1: (No Response)

Reviewer #2: (No Response)

2. Is the manuscript technically sound, and do the data support the conclusions?

Reviewer #1: Yes

Reviewer #2: Yes

3. Has the statistical analysis been performed appropriately and rigorously? 

Reviewer #1: Yes

Reviewer #2: Yes

4. Have the authors made all data underlying the findings in their manuscript fully available?

Reviewer #1: (No Response)

Reviewer #2: No

5. Is the manuscript presented in an intelligible fashion and written in standard English?

Reviewer #1: Yes

Reviewer #2: Yes

6. Review Comments to the Author

Reviewer #1: (No Response)

Reviewer #2: Review of the manuscript titled “Static loading of the knee joint results in modified single leg landing biomechanics”. (PONE-D-19-17819)

Dear Dr. Mirkov

I would like to thank you, once again, for giving me the opportunity to review the manuscript titled “Static loading of the knee joint results in modified single leg landing biomechanics”. This is undoubtedly a very important work the results of which will help both researchers and clinical therapists to better understand the mechanism of knee injuries. Unfortunately, many times the reviewer is obliged to identify ambiguities that are clear in the researcher's mind, only to improve the image of the manuscript to third parties and not to question the value of the work. This is often a demanding task and often frustrates the researcher as the project's publication is delayed.

The main concern of the last revision was the clarification of the terms "pre-landing" and "post-landing" conditions as well as the terms "pre-loading" and "post-loading" conditions which were used alternatively throughout the manuscript. Although it was clear to me that comparisons were made before and after passive knee loading, that is “pre- and post-loading” of the knee, this was not consistently reported (as in the case of ground reaction forces-GRF in both text and image), confusing the reader. However, the explanatory terms given by the author are sufficient and understandable. In the spirit of the above comments, the author should consider modifying the point that was addressed in the previous revision and although it was claimed that was modified it is not clear to me if this was actually happened. The point I am referring to is in the Abstract Page 2, Line 38: What "a" stands for in the abbreviation "aVGRF"? and in Line 39: What NAEMG stand for? Is it different from the NEMG that has been modified and used throughout the manuscript?

Furthermore the “Overall Average and Maximal NEMG” data that I was referring to, although it was removed from the Table in a previous modification of the manuscript, it remains as a subtitle of the entire paragraph that is dedicated to the results obtained from the analysis of EMG signals. Although its correctness/accuracy was not questioned the author should consider, in the final version of the manuscript, to modify the subtitle since this paragraph contains multiple comparison of the EMG data and not just the “Overall”.

Finally, the negative signs, in Table 3 have been clarified only for “Ankle flexion”. The negative signs for “Knee abduction” and “Hip abduction” have not yet been clarified (as proposed in the previous revision). Since the author clarify the negative signs in Table 4 for angular velocity (“…are indicative of positive and negative rates of movement, respectively”) he may consider adding the information as a footnote in the Table.

Sincerely

7. PLOS authors have the option to publish the peer review history of their article (what does this mean?). If published, this will include your full peer review and any attached files.

Reviewer #1: No

Reviewer #2: Yes: Dimitris Mandalidis, Assistant Professor, Department of Physical Education and Sports Science, National and Kapodistrian University of Athens

---

## [Editor Report · Acceptance letter]

7 Feb 2020

PONE-D-19-17819R2 

Static loading of the knee joint results in modified single leg landing biomechanics 

Dear Dr. Olson:

I am pleased to inform you that your manuscript has been deemed suitable for publication in PLOS ONE. Congratulations! Your manuscript is now with our production department. 

With kind regards,

on behalf of

Dr. Dragan Mirkov 

Academic Editor

PLOS ONE